# Feasibility and acceptability of continuous glucose monitoring in pregnancy for the diagnosis of gestational diabetes: A single-centre prospective mixed methods study

Laura C. Kusinski[1], Joanne Brown[1,2], Deborah J. Hughes[1,2], Claire L. Meek[1,2]*

1 Wellcome Trust–MRC Institute of Metabolic Science, University of Cambridge, Cambridge, United Kingdom, 2 Cambridge Universities NHS Foundation Trust, Cambridge, United Kingdom

* clm70@cam.ac.uk

## Abstract

### Background

Undiagnosed diabetes in pregnancy is associated with stillbirth and perinatal complications, but standard testing for gestational diabetes using the oral glucose tolerance test (OGTT) is impractical and exacerbates healthcare inequalities. There is an urgent need to improve the accuracy, acceptability and accessibility of glucose testing in pregnancy. We qualitatively assessed the feasibility and acceptability of two alternative home-based methods of glucose testing in pregnant women, using continuous glucose monitoring (CGM), with or without a home-based OGTT.

### Methods

We recruited women with a singleton pregnancy at 28 weeks' gestation with $\geq 1$ risk factor for gestational diabetes attending antenatal glucose testing. A Dexcom G6 CGM device was sited and women were asked to take a 75g OGTT solution (Rapilose) on day 4 after an overnight fast. Qualitative interviews were performed with 20 participants using video conferencing according to a semi-structured interview schedule and thematically analysed using NVIVO software.

### Results

92 women were recruited; 73 also underwent a home OGTT. Women had an average of 6.9 days of glucose monitoring and found the CGM painless, easy to use with few or no adverse events. During the qualitative study, the main themes identified were reassurance and convenience. All women interviewed would recommend CGM and a home OGTT for diagnosis of gestational diabetes.

### Conclusions

CGM with or without a home OGTT is feasible and acceptable to pregnant women for diagnosis of gestational diabetes and offered advantages of convenience and reassurance.

**Data Availability Statement:** All relevant data are within the manuscript and its Supporting Information files.

**Funding:** This study was funded by Diabetes UK through an intermediate clinical fellowship to CLM (17/0005712; ISRCTN number 90795724). CLM is also supported by the EFSD-Novo Nordisk Foundation Future Leader's Award (NNF19SA058974). Dexcom supplied the CGM systems and some research nurse staff costs to support patient care during the pandemic. The funders had no role in study design, data collection and analysis, decision to publish, or preparation of the manuscript.

**Competing interests:** Dexcom supplied the G6 equipment and a grant for staff time for this investigator-initiated project (AR21771).The authors have declared that no other competing interests exist. This does not alter our adherence to PLOS ONE policies on sharing data and materials.

Further work is needed to clarify diagnostic thresholds for gestational diabetes using CGM metrics.

## Introduction

Hyperglycaemia in pregnancy affects 1 in 6 pregnancies internationally and is associated with perinatal complications affecting mother and child [1]. Accurate diagnosis of gestational diabetes, the most common form of hyperglycaemia in pregnancy, is essential to facilitate optimal clinical management and reduce adverse events associated with undiagnosed or untreated gestational diabetes, such as stillbirth, neonatal death and birth injury [2–5]. However, the optimal method for screening and diagnosis of gestational diabetes remains unclear and is a topic of intense controversy internationally.

The oral glucose tolerance test (OGTT) is considered the "gold standard" test for gestational diabetes, which involves patients having venous blood sampling at timed intervals (0, 1 and 2 hours) after consuming a standard 75g glucose drink after an overnight fast [6]. This method, although being the gold standard, has issues of inaccuracy, impracticality and poor tolerability among pregnant women [7]. Individuals having two OGTTs within a week will only receive the same result on around 27–80% of occasions[8–10] and fasting values can vary by 10–30% [8]. This poor diagnostic performance may be even worse in pregnancy, where a greater degree of variability has been reported [11]. Our previous work suggested that the OGTT exacerbates inequalities in healthcare delivery and outcomes, as it is particularly unappealing to women with important risk factors (ethnicity, obesity, family history of diabetes and socioeconomic disadvantage) who choose not to attend [7].

A further concern relates to the practicality of OGTTs: they require two hours of patient and staff time which is expensive and inconvenient. The requirement for attendance in the morning can also be challenging around childcare commitments. Capacity for OGTT testing is limited, but as population rates of obesity rise, more OGTT appointments will be needed.

There is therefore an urgent need to identify robust diagnostic strategies for gestational diabetes which can be performed outside of hospital and reduce burden on both patients and healthcare services. Continuous glucose monitoring (CGM) is a well-established technology using a subcutaneous sensor to measure glucose concentrations in interstitial fluid every five minutes for 7–10 days [12]. CGM is generally well-tolerated by people with type 1 and type 2 diabetes [13–15] and considered acceptable and beneficial by expert users [16,17]. In addition, CGM is increasingly used in pregnant women with type 1 diabetes as it has been found to improve pregnancy outcomes [18], but it is less commonly used in type 2 diabetes in pregnancy, gestational diabetes or in pregnant women without a diagnosis of diabetes. The acceptability of CGM has been reported favourably in populations of women with type 2 diabetes in pregnancy [19] and has been under increasing scrutiny in gestational diabetes. CGM offers a novel approach to improve the diagnosis of gestational diabetes and could potentially also improve maternal glycaemic control and pregnancy outcomes. A pilot study performed by Filippo and colleagues suggested that CGM represents a more acceptable alternative for gestational diabetes diagnosis to an OGTT [20] which was confirmed by a subsequent pilot study by the same researchers [21]. However, although preliminary work suggests that CGM can improve maternal glycaemia and reduce offspring birthweight, further studies are needed to confirm the clinical value of CGM in gestational diabetes management [22].

In this study, we focus on the potential of CGM to improve the diagnosis of gestational diabetes through two novel approaches. Firstly, CGM metrics in common use [12] may be just as

predictive of pregnancy outcomes as the standard venous OGTT, or possibly more predictive, as it provides glucose information over a longer period of time. Secondly, CGM provides the opportunity to perform a home-based OGTT using measures of CGM glucose instead of venous glucose. The aim of this study was to qualitatively assess the feasibility and acceptability of CGM use in pregnancy and its use in a home-based OGTT for diagnosis of gestational diabetes.

## Participants and methods

### Study ethical considerations

The study was approved by the United Kingdom Health Research Authority and East Midlands Research Ethics committee (20/EM/0133; IRAS 282553). All participants provided written informed consent.

### Study design and participants

This single-centre, prospective, observational study included pregnant women who were attending Cambridge University Hospitals NHS Foundation Trust for antenatal glucose testing between June 2020 and December 2021. Pregnant women were eligible if they had a live singleton fetus confirmed by ultrasound, who were being screened for gestational diabetes at around 28 weeks of pregnancy with one or more risk factors for hyperglycaemia (criteria of the National Institute for Health and Care Excellence [23]. At this time during the COVID-19 epidemic, women could only come to the hospital alone for scheduled appointments, and standard diagnostic procedures using a hospital OGTT were not performed [23]. Potential participants received information about the study before their blood test appointment and if interested, discussed the study on the telephone with a research nurse. Participants who gave verbal consent were invited to attend the phlebotomy department where they had blood taken as part of their standard care (fasting or random glucose and HbA1c) in addition to a short research visit. For the research study, consent was confirmed in writing, and a CGM sensor (Dexcom G6; Dexcom Ltd, San Diego, United States) was sited in their upper arm and masked to ensure participants could not see their glucose results in real time. The participants were given a CGM receiver and brief verbal and written instructions about how to use it (for example, to keep the receiver within 6m of the sensor at all times). Participants were not reimbursed for their time.

### Exclusion criteria

Women were excluded from the study if they had any of the following conditions; multiple pregnancy; pregnancy with confirmed or high clinical suspicion of congenital anomaly, major active concurrent physical or psychiatric illness, such as symptomatic cardiac disease; kidney failure requiring dialysis or previous organ transplantation; taking medication such as oral corticosteroids (not inhaled), atypical antipsychotics and antiretroviral medications, or other medications known to interfere with glucose homeostasis.

### Home OGTT

For the home OGTT, participants were given a sachet of Rapilose 75g OGTT (Galen Ltd, Craigavon, UK) solution to take home with strict instructions of how to perform a home OGTT (see S2 File). On day 2 or 3 after CGM insertion, participants were asked to fast overnight for at least 10 hours and drink the OGTT solution the following morning (day 3 or 4) at 09.00hrs exactly, and to consume the whole drink within 5 minutes. Women were asked to

**Table 1. Baseline characteristics of women who participated in the study.**

| | Pregnant women at risk of gestational diabetes (GestTesting study) N = 92 | Pregnant women who participated in the qualitative interviews N = 20 |
|---|---|---|
| **Baseline Characteristics** | | |
| Age at recruitment years | 33.3 (4.9) | 32.0 (4.0) |
| BMI at recruitment kg/m$^2$ | 31.3 (6.8) | 30.8 (7.2) |
| Ethnicity | White British 54/92 (58.7%) White Other 10/92 (10.8%) Asian 17/92 (18.5%) Black/African 3/92 (3.3%) Chinese 2/92 (2.2%) White/Asian 1/92 (1.1%) White/Black Caribbean 5/92 (5.4%) | White British 10/20 (50.0%) White Other 4/20 (20.0%) Asian 4/20 (20.0%) Black/African 1/20 (5.0%) Chinese 1/20 (5.0%) White/Asian 0/20 (0%) White/Black Caribbean 0/20 (0%) |
| Primiparous | 44/92 (47.8%) | 11/20 (55.0%) |
| Gestational age at recruitment weeks | 27.8 (1.75) | 28.1 (0.38) |
| HbA1c mmol/mol | 30.7 (3.1) | 30.4 (2.9) |
| HbA1c % | 5.0 (0.3) | 4.9 (0.3) |

Results are presented as mean (SD) or n (%).

document the exact time the OGTT was started and a reminder was sent the day before via text message. Women could choose to do this on day 3 or 4 to allow a little flexibility around other commitments, while ensuring the OGTT timing coincided with peak sensor accuracy [24]. Women were instructed to not eat for 3 hours after the ingestion of the Rapilose solution. The CGM sensor recordings at time 0, 60 and 120 minutes were used in analysis. After 7–10 days, the sensor stops transmitting data. Participants removed the sensor themselves at home and placed it in a plastic pouch for return to the study team. The data from the CGM was uploaded using Dexcom Clarity software. Data are presented as n (%) or mean (SD) in Table 1.

## Gestational diabetes diagnosis

Women in the study were diagnosed with gestational diabetes using the novel testing strategy which became standard care during the Covid-19 pandemic [25] (HbA1C $\geq$ 39 mmol/mol or Random plasma glucose >9 mmol/L or Fasting plasma glucose $\geq$ 5.6 mmol/l). Prior to the Covid-19 pandemic, women would be diagnosed using the World Health Organisation guidelines using a 75g oral glucose tolerance test with glucose concentrations meeting one of the following criteria; fasting plasma glucose level of 5.1 mmol/l or above; 1-hour plasma glucose level of 10.0 mmol/l or more or a 2-hour plasma glucose level of 8.5 mmol/l or above [26]. The use of the CGM data was not routinely made available to the clinical team for clinical decision making unless there was a clear safety risk to the participant or her baby (Thresholds used to identify risks were as follows: Mean glucose of >8mmol/l or time in range <60%). Women considered intermediate in risk, with CGM glucose results at 0, 60 or 120 minutes which were consistently above the diagnostic thresholds used in our centre (criteria of the International Association of the Diabetes in Pregnancy Study Groups; fasting glucose $\geq$5.1 mmol/l; 60-minute glucose $\geq$10.0 mmol/l; 120-minute glucose $\geq$8.5 mmol/l), or who had CGM Mean glucose of >7mmol/l or time in range <80%, features of macrosomia (estimated fetal weight >90[th] centile) or symptoms of hyperglycaemia were referred for a 1-week period of glucometer monitoring and a medical review (by CLM).

## Qualitative analysis

To assess the feasibility of using CGM more widely for gestational diabetes diagnosis, participants were asked if they would be willing to take part in a qualitative interview. A voluntary response sampling method was utilised, and consenting women were contacted via video conferencing software and audio-recorded verbatim (n = 20). A study code was used for all recordings and transcripts were checked for anonymity. Non-identifiable transcripts of interviews are available to readers upon written request to the corresponding author. The semi-structured interview questions to be used for the qualitative aspect of the study are to be found in the (S1 File). The qualitative interviewer (LCK) is a post-doctoral research associate (PhD) in diabetes in pregnancy and was trained in qualitative research. Interviews were designed to understand the participants overall experience of taking part in the study.

During the interview women were given the opportunity to ask any additional questions and give feedback which was not covered by the questions. Recorded interviews were transcribed by an external company (The Typing works, Pinner) and analysed following the Braun and Clarke six step process [27] using NVIVO software (Timberlake, UK). Initial analysis provided a set of codes drawn from the interview questions which detailed the main themes in the data set. After all transcripts were coded, main themes were identified, and data reviewed to finalise subthemes. With the use of the dynamic software NVivo, further data analysis was performed to enable comparisons between participants. The overall data was able to provide a clear narrative-based account of individual experiences but also able to identify common themes allowing for a more generalised view of the intervention. We used the framework of acceptability described by Sekhon and colleagues to determine acceptability [28]. This consists of seven component constructs: affective attitude, burden, perceived effectiveness, ethicality, intervention coherence, opportunity costs, and self-efficacy. We considered that the perceived effectiveness, ethicality, intervention coherence, opportunity costs, and self-efficacy could not be fully assessed by this study, as the clinical benefits of using CGM for diagnosis or management of gestational diabetes are unknown. We therefore focussed this mixed methods study on questions about affective attitude and burden.

## Results

Between June 2020 and December 2021, 92 women were recruited and wore a masked CGM for an average of 6.9 (1.7) days. The majority of women (73/92; 79%) completed an at home OGTT. The remaining women did not complete the test due to illness during the week of the test (n = 1), device failure (n = 1), OGTT had been reintroduced to the clinic setting (n = 10) or personal preference to avoid the glucose drink (n = 7). A minority of women (2/92; 2%) had gestational diabetes diagnosed using the Covid-19 criteria [25]. Baseline characteristics and HbA1c data are presented in Table 1.

To assess if the clinical feasibility of the home OGTT was sufficient, we assessed women's adherence to the written instructions. The majority of women (60/73; 82%) reported consuming the Rapilose within 5 minutes of 9.00hrs, as requested. Among the remaining women, (9/73; 12%) reported not drinking the Rapilose within 5 minutes, (2/73; 3%) had CGM failure and (2/73; 3%) did not give details of whether they drank it in the allocated time.

## Qualitative themes

Twenty participants took part in the qualitative interviews who were statistically similar to the cohort as a whole. After twenty participants were interviewed, we were no longer identifying new themes within the data and thus considered that data saturation had been achieved. Two

major themes were detected in the participant feedback and these were reassurance and convenience. These will both be considered in turn with the relevant subthemes also described.

Affective Attitude: Reassurance

## Importance of glucose testing in pregnancy

This subtheme addressed women's perceptions of the importance of glucose testing in pregnancy for their health and the health of their baby. Women were generally concerned about the new diagnostic process for gestational diabetes adopted during the Covid-19 pandemic and were worried the testing procedure was not accurate enough to diagnose the condition. Almost all women were aware of the importance of antenatal glucose testing and the impact that poor glucose control could have on their baby (additional quotes related to reassurance and the subthemes can be viewed in Table 2).

*"I think very nervous and very worried, because I kind of wondered if it would be sufficient to pick up if I did have gestational diabetes during pregnancy, so I wanted something a little bit more kind of in depth, something a little bit more robust"*

**Table 2. Participant quotes about the CGM and home OGTT relating to the main theme of reassurance.**

| Sub theme | Participant's Quote |
| --- | --- |
| **Importance of glucose testing in pregnancy** | "Due to my previous two pregnancies and having large babies and I wasn't given a Glucose Tolerance Test at 28 weeks, and so when I had my last pregnancy and birth it was really traumatic. So I wanted a GTT in this pregnancy so they should have screened but didn't because my sister's a Type 1 Diabetic, so I know a fair bit about it. And then the coronavirus obviously started and then they stopped the Glucose Tolerance Test which worried me" |
|  | "I'm obviously aware that it's, you know, it's a risk during pregnancy. And, I think, because of my BMI I was probably going to be, supposed to be tested for it, anyway, so, yeah. Also, because I have a chocolate habit, so a bit paranoid that I've given myself gestational diabetes." |
| **Limiting time in hospital** | "Yeah, I would actually because it was really quick and easy way of doing it. So I'd say that's probably a good, rather than having to wait around the hospital and wait two hours, it was a really good way of doing it I suppose." |
|  | "I think like in this, like in the case of during this pandemic time and everything I think this is much convenient than the people who are not working, or who are not in direct contact with their hospital, or some of the people who may be needing shielding and everything, so for that, people I think, like glucose monitoring at home will be much better than sitting at the hospital." |
|  | "Yes, I mean previously when I've had glucose testing done with my older daughter, you used to have to sort of wait around in the hospital to get results back, so it was quite nice to actually have the little chip put in and then leave and do it from home, so I thought that was actually quite nice rather than sitting around in a hospital. And at the moment obviously no-one wants to be doing that because of Covid and I know obviously for yours and your colleague's point of view it's safer not to be there and totally understandable, so yes that was absolutely fine and worked quite well actually." |
|  | "Definitely, I can imagine, yeah, I can imagine that compared to having to sit in the hospital for hours, it's much nicer just to be at home and just to be in the comfort of my own home to do it. I also felt quite reassured by the fact that my blood sugars were being monitored for a whole week, and I thought that, that you would have more data to look at. So obviously, I mean, depending on the results of the study, but I, my guess is that, I would have thought it was quite a good way of monitoring people" |
| **Comfort of doing in a home setting** | "it's quite nice where you can sort of do it from the comfort of your own home and it's, you know, quite a relaxed sort of routine, so yeah I couldn't see any reason why not, I would recommend it, yes." |

The additional testing with the CGM in this study provided extra reassurance to the women, that they were being monitored for gestational diabetes.

### Burden: Limiting time in hospital

Most women were glad that they did not have to go into hospital and spend a long period of time waiting while they had a hospital OGTT but were unsure that the HbA1c or random /fasting glucose tests alone would be enough to recognise whether or not they had gestational diabetes. The opportunity for them to have their glucose monitored over a longer period of time in the study gave them extra reassurance that they were being assessed more thoroughly.

*"I know you have to wait like three hours in the hospital before and I know that nobody would want to do that now and I wouldn't feel comfortable doing that now so I understood why you've been looking for other ways to do the testing."*

### Burden: Comfort of testing in a home setting

The home OGTT was more comfortable to participants and all were happy with doing this in a home setting.

*"Personally I feel like I, obviously I haven't done it before but I feel like I would have been a lot more nervous doing it in the hospital anyway like before Covid, I much preferred doing it at home because I'd heard people saying it's not a nice drink and that they'd be sick after or things like that so that's, it was quite nice to know that if I was, I was at home if that makes sense I felt a lot, yes like more comfortable doing it at home to be which was a really good thing"*

Overall the reassurance of longer-term CGM glucose monitoring and the additional home OGTT was considered a major advantage of this study and all women were grateful for this extra surveillance.
Burden: Convenience

### Burden: Limiting hospital trips

This subtheme identified aspects of convenience with a home OGTT compared to the hospital OGTT. Women considered the study, with CGM and a home OGTT to be convenient. The single appointment with blood drawn for laboratory analysis and initiating the CGM sensor session was viewed as convenient and appealing to the women in this study. (Additional quotes related to convenience can be viewed in Table 3).

*"I welcomed the fact that I could do it at home and I, my trip to the hospital just coincided with me having my routine 28-week blood test so I was there anyway when the device was fitted, so it was, I didn't make any extra trips or anything to have to have it done."*

### Burden: Minimal work/life disruption with home OGTT

This subtheme identified aspects of convenience related to work-life disruption. The convenience of study was praised further as the home OGTT meant that again there were no additional trips to the hospital required and that this test could be done in the comfort of their own home. Often women had busy working schedules and would have felt inconvenienced if they

**Table 3. Participant quotes about the CGM and home OGTT relating to the main theme of convenience.**

| Sub theme | Participants Quote |
|---|---|
| **Limiting hospital trips** | "it fitted in quite well really because I, I did and I didn't really have that amount of time spare to just sit in hospital but also you don't really want to sit in hospital at this time so I was quite pleased that she said you literally come in for, I think I was all-in-all in hospital for less than half an hour for them to fit it in my arm and I did have to have other bloods taken so if you actually subtract the, the period of time where I had to have my other bloods taken I was in hospital for less than 20 minutes, it was injected in my arm and I left. It was easier, it's faster, it's less invasive." |
| **Minimal work/life disruption with home OGTT** | "I think staying at home was quite helpful because normally if I'm working I leave home about 7-ish, so that would have been, and then I'm in the appointments throughout the day so I wouldn't have time to, unless I'm going to hospital and made a time specially for it, then I probably would have missed the drink or I would have completely missed the timing, taking it nine o'clock, finishing it within the three minutes and then starting the food kind of thing. So I probably would have missed it, but being at home was quite easy." |
| **Comfort of wearing the CGM** | "when she put it on my arm I didn't really feel anything and during the whole week it was, I forgot about it so it was very good."<br><br>"it was fine, as it was a very small thing and it was not that painful while inserting also and the sticky part was quite sticking really well, so it was, really stick good to the arm. There was no issues with me while using that monitor." |
| **Inconvenience of CGM monitor receiver handling** | "It was actually, it was really, really good and I had it by me most of the time. The only thing that I found difficult was if I was working at the office I would tend to sort of run downstairs or something like that and completely forget that it was in my, outside my range. So that was the only problem I had with it, just forgetting that to be in the same distance of it, or I'd put it in my bag to make sure it was near me because I carry my bag with me everywhere, and then when I got home, I'd go and sit on the sofa and it'd be in the loun. . . in the kitchen. So I think that was the only sort of drawback with it." |
| **Recommendation to other pregnant women at risk of gestational diabetes** | "Yes because I say, I am, I speak to a lot of pregnant women on my Facebook groups and I know that they're a bit wary of going into hospitals and having, and obviously wearing a mask for three hours would be very uncomfortable so I think doing it at home, it makes you feel a bit safer and you say you don't even think about it to be honest." |
| | "I would do again, definitely, because if I can help more pregnant women in the future, that's, I think that's the goal. The point is to get better and it's nothing that affects the baby, it's really only in me. So I'm happy, because basically it's a win/win situation, you give me extra care, let's say during the Covid situation, and at the same time I give back to society, so if that's not win/win, then what? So, yes, I would definitely do it again or any other research that doesn't cause any harm, to me or the baby, yes." |
| | "Yeah, absolutely, without a doubt, yeah. I think it makes more, not it makes more sense, but I think the ease of taking part is just way better than three hours or two hours at Addenbrooke's, which was, I guess, the old method. But, yeah, no, this is way, it's a lot more convenient than, like I said, you know, I had it on for seven, eight days and I didn't even notice at all that it was on, so it wasn't really a big inconvenience." |

would have had to take time off to come to the hospital to do an OGTT. Being able to do glucose testing at home meant they could continue with their daily work or routine at home without much interruption.

> "I do believe that this testing is very important, but it's almost 3, 3 hours of your life that you just sit in hospital, it's not, it's a, it's like a redundant time, when you could be at home doing the fast exactly the same, getting on with your life or going to work, I wasn't at work but you could be doing everything normal whilst doing this test rather than having to take a half day holiday from work or anything like that. So in my eyes yes I think it's very good."

## Burden: Comfort of wearing the CGM

This subtheme addressed aspects of comfort of CGM–both in relation to the placement of the sensor and the comfort of wearing it for multiple days. Participants were pleased that placement of the CGM sensor was easy and painless. Many commented on how quickly they forgot about having it in their arm.

> "I was pleasantly surprised because I couldn't feel it at all. So like the actual monitor in my arm didn't bother me at all, I basically forgot it was there most of the time"

> "it didn't hurt and it was absolutely fine. There was a few times I was a bit worried about catching it, you know, sort of in day-to-day life and things like that but it, actually I forgot that it was there after a couple of days and it wasn't too bad."

Some participants struggled with having to remember to keep the CGM receiver in close proximity with them to ensure that signal was not lost. This was not a major issue but caused minor frustration, as women were not used to having to carry something additional on their person.

> "I think the only thing kind of that you had to get used to it was that you had to have it with you within the six metres, and that was the only thing pretty much, so I think if it was kind of like you just had to have it constantly on you just to make sure that it obviously doesn't lose its signal."

The only side effect mentioned by some participants was that the CGM made their arm itchy, but this would not put them off recommending doing it to other ladies.

> "my skin's fine, it didn't leave any marks or anything, I think it just got itchy because it was on for so long."

> "Quite happy with it, although sometimes the thing on my arm like make me feel a little bit itchy but it's not really a big problem, just sometimes I can feel it, but overall I think it was quite good."

## Overall acceptability: Recommendation to other pregnant women at risk of gestational diabetes

From the interviews it was clear that overall women were very pleased with the convenience of the study and were happy that they had taken part. This was emphasised when asked a final question regarding whether they would recommend taking part to other women who were at

risk of having gestational diabetes. All 20 women interviewed confirmed that they would recommend it to other pregnant women.

> *"overall for me it was a very, very pleasant experience."*

> *"Definitely, 100%, for sure [laughs] I don't think I can recommend it enough. And the main reason, I think, is that it's very relaxing the way it's done, because I didn't have to go to the hospital, which doesn't sound, I mean, it's nothing negative, but just the idea of going in and staying in for three hours and having your blood taken is more stressful comparing to being in your own house and having the sensor on, which, as I said before, I didn't even notice it. Yes, so I would. Now, for me it wasn't stressful at all. I don't know if, for a woman, if it's easier to just have one test done in one day and then that's it. But for me, being monitored for a period of seven to ten days felt much more reassuring comparing to just one test on one day."*

Home-based glucose testing with CGM, with or without an OGTT, was therefore convenient, appealing and acceptable to women at risk of gestational diabetes during the Covid-19 pandemic.

## Discussion

This study demonstrates that CGM is acceptable to pregnant women as an alternative means of testing for gestational diabetes, and offered advantages, particularly convenience (reduced burden) and reassurance (positive affective attitude) during the Covid-19 pandemic.

This study has a number of strengths. We qualitatively evaluated a novel diagnostic process for gestational diabetes, using CGM as a diagnostic test, rather than a tool for self-monitoring in established diabetes. We used CGM in pregnant women without known diabetes at the time of testing, demonstrating its widespread acceptability even in a healthy population. We assessed the feasibility and acceptability of home-testing with CGM and an OGTT in women at high risk of gestational diabetes from a range of ethnic backgrounds. Despite the challenges of conducting clinical research during the Covid-19 pandemic, we found participants were very willing to engage in qualitative interviews over video conferencing facilities.

This study has some weaknesses. While we approached all eligible women, it is possible that women who agreed to participate were potentially more concerned about gestational diabetes, or more aware of antenatal risks. This group of women were more likely to find this study useful and adhere to the study procedures carefully. On account of the pandemic, we were unable to assess the comparative accuracy of CGM compared to venous or capillary glucose testing, but this data is available elsewhere [29–31].

Importantly, this study is also the first to qualitatively assess women's attitudes to the new testing process which was introduced during the Covid-19 pandemic. We identified that women were concerned with the limited testing that was available for gestational diabetes and were anxious that diagnosis may be missed. The women interviewed in this study found the extra testing with the CGM and home OGTT provided extra reassurance that they were undergoing extra surveillance. Women welcomed the possibility that this method may give them a more accurate gestational diabetes diagnosis.

Performing an OGTT at home raises a number of challenges affecting the diagnostic validity of the test. Women need to take the glucose solution on time after an appropriate duration of fasting. The timing of the glucose ingestion must be known exactly in order to allow for accurate timing of the 0, 60 and 120-minute CGM readings. Women are not supervised and therefore we cannot exclude the possibility that women had insufficient duration of fasting, excessive activity during the test or consumed insufficient amounts of the Rapilose solution.

However, standard hospital OGTTs also have unmeasured factors which affect their validity. In clinical care, even with some supervision, we cannot exclude insufficient glucose ingestion, insufficient fasting or excessive activity in all cases. Furthermore, depending upon service capacity, hospital OGTTs can start from 07.00 to 10.00hrs, a difference which is likely to give quite different fasting readings. Timing of the 1- and 2-hour values can be subject to delays in a busy department. Hospital glucose testing is also subject to inaccuracies due to pre-analytical processing and delayed transportation to the laboratory. These differences are known to give profound differences in OGTT performance and gestational diabetes diagnosis rates in clinical care.

The use of CGM as a novel method of diagnosing gestational diabetes therefore had some advantages during the Covid-19 pandemic and holds promise for more widespread use in the longer-term. Compared to a hospital OGTT and random/fasting glucose tests, CGM is less influenced by pre-analytical processing inaccuracies. Unlike HbA1c, CGM glucose measures are not influenced by red cell turnover or iron deficiency, and may give more consistent glycaemic information in pregnant women regardless of individual haematological parameters. Using CGM for glucose testing in pregnancy also provides an opportunity to reduce healthcare inequalities, as women from minoritised or disadvantaged groups find the OGTT particularly unappealing [7].

As CGM devices improve in accuracy, diagnostic testing for gestational diabetes using CGM may offer increased testing capacity, and increased convenience both to women and healthcare services at a comparable cost to a hospital OGTT. This study demonstrates that women found the CGM to be a positive experience as a whole: easy to use, painless at insertion and providing extra reassurance. All 20 women would recommend this as a technique for future testing for gestational diabetes. We consider out-of-hospital testing to be a good option for women in the UK and internationally, especially where access to formal healthcare settings is difficult. This shows that this alternative process for gestational diabetes diagnosis may be a viable option for future clinical use. This is in line with results seen by Filippo and colleagues, who used a different methodological approach in pregnant women and healthcare professionals using a custom-designed questionnaire on CGM acceptability [20]. The authors identified that women found CGM to be significantly more acceptable that OGTT and ninety-three percent of the participants would recommend it for diagnosis of gestational diabetes[20]. In our study, all women recommended using CGM. Taken together, these studies suggest that CGM is associated with very high levels of acceptability even in pregnant women without a diagnosis of diabetes.

In conclusion, we demonstrate that home-based glucose testing using CGM is feasible, convenient and acceptable to pregnant women as an alternative method for the diagnosis of gestational diabetes. Future work will assess the optimal CGM metrics to identify women with gestational diabetes and predict suboptimal pregnancy outcomes.

## Supporting information

**S1 File. Qualitative interview questions asked to the participants after they had completed the study.**
(DOCX)

**S2 File. Instructions given about the OGTT.**
(DOCX)

## Acknowledgments

We thank our participants for their willingness to take part in this study. Dexcom, Inc. kindly supplied free Dexcom G6 CGM systems and supported research nurse staff costs to improve

diabetes care during the Covid-19 pandemic. We are grateful to the staff in Cambridge University Hospitals NHS Foundation Trust, including the phlebotomy department and diabetes in pregnancy service, and the National Institute of Health Research (NIHR) Clinical Research Network (CRN Eastern) for supporting this study.

## Author Contributions

**Conceptualization:** Laura C. Kusinski, Claire L. Meek.

**Data curation:** Laura C. Kusinski, Claire L. Meek.

**Formal analysis:** Laura C. Kusinski, Claire L. Meek.

**Funding acquisition:** Claire L. Meek.

**Investigation:** Laura C. Kusinski, Claire L. Meek.

**Methodology:** Laura C. Kusinski, Claire L. Meek.

**Project administration:** Laura C. Kusinski, Joanne Brown, Deborah J. Hughes.

**Resources:** Claire L. Meek.

**Supervision:** Claire L. Meek.

**Validation:** Laura C. Kusinski.

**Writing – original draft:** Laura C. Kusinski.

**Writing – review & editing:** Laura C. Kusinski, Joanne Brown, Deborah J. Hughes, Claire L. Meek.

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
