## [Decision Letter · Decision Letter 0]

6 Jun 2023

PONE-D-23-07848Feasibility and acceptability of continuous glucose monitoring in pregnancy for the diagnosis of gestational diabetesPLOS ONE

Dear Dr. Meek,

Thank you for submitting your manuscript to PLOS ONE. After careful consideration, we feel that it has merit but does not fully meet PLOS ONE’s publication criteria as it currently stands. Therefore, we invite you to submit a revised version of the manuscript that addresses the points raised during the review process.

We look forward to receiving your revised manuscript.

Kind regards,

Graeme Hoddinott, Ph.D

Academic Editor

PLOS ONE

Journal Requirements:

"Dexcom supplied the G6 equipment and a grant for staff time for this investigator-initiated project (AR21771).The authors have declared that no other competing interests exist." 

**Additional Editor Comments:**

Please address or rebut the reviewers' comments. These align for the most part and in general are asking for additional clarity. 

Reviewers' comments:

Reviewer's Responses to Questions

**Comments to the Author**

1. Is the manuscript technically sound, and do the data support the conclusions?

Reviewer #1: Partly

Reviewer #2: Partly

2. Has the statistical analysis been performed appropriately and rigorously? 

Reviewer #1: Yes

Reviewer #2: I Don't Know

3. Have the authors made all data underlying the findings in their manuscript fully available?

Reviewer #1: No

Reviewer #2: No

4. Is the manuscript presented in an intelligible fashion and written in standard English?

Reviewer #1: Yes

Reviewer #2: Yes

5. Review Comments to the Author

Reviewer #1: Thank you for doing this research on a significant issue. The use of CGM seems to have a powerful future role in both the diagnosis and the management of GDM.

Comments:

1. There are very few studies on the use of CGM in the diagnosis, which also report their acceptability among women. Please, cite them in the introduction and also make a comparison with your study in the discussion.

2. Also, Can you please describe the clinical/ scientific reasoning for masking the CGM data from the clinical decision team?

3. There have been separate questions specifically asked to assess whether the process of wearing or removing the device was found to be painful or not. I understand that one participant mentioned it, but that is very important when we assess acceptability. Also, documentation on any side effects.

Reviewer #2: General comment: This manuscript makes an important contribution to the field of gestational diabetes; suggesting that CGM (potentially combined with OGTT) done at home, is more acceptable than standard of care. However, I have a few concerns about the qualitative aspect of the research. I make a few suggestions and raise a few minor questions below.

Title:

Suggest including setting or location in which study was conducted in the title.

Abstract:

Line 33: 'pre-determined interview schedule' is later referred to as 'semi-structured interview questions'. Suggest using consistent terminology throughout. The former seems more in line with a 'structured interview schedule' than a semi-structured interview schedule. This has methodological implications.

Introduction:

Line 53: Suggest including a brief description of what OGTT entails i.e., in standard of care, to better orient the reader to the following two paragraphs.

Line 60: remove 'high risk' from 'high risk ethnicity'. Ethnicity is a risk factor.

Line 75: remove 'the' before 'CGM'.

Participants and methods:

Since these data were collected during the peak of the COVID-19 pandemic, it might be helpful to include a brief description of the setting in which the data were collected. E.g., what, if any, restrictions were in place regarding healthcare access?

Study design and participants:

Line 84: The current description of the study design, only describes the 'clinical' side of the study. It does not describe the design of the qualitative study.

Lines 84-86: Suggest this be moved to a separate section entitled, 'ethics' or 'ethical considerations'. In this section, also include whether participants were reimbursed.

Recruitment process:

There is no description of how the 20 participants recruited for the qualitative part of the study were selected. What sampling approach was followed?

Line 92: 'pregnancy with one of more risk factors', do you mean 'pregnancy with one or more risk factors'?

GDM diagnosis:

Lines 126-127: How does this differ from the diagnostic and processual criteria followed outside of COVID-19 conditions?

Qualitative analysis:

There are two sections in the manuscript entitled 'qualitative analysis'. (See Line 139 and Line 166).

Why were only 20 participants recruited to the qualitative study? Was saturation reached?

Lines 147-149: What kind of thematic analysis was used? Provide references for the approach used. E.g., What framework of acceptability was used to guide analysis?

Results:

Line 159: In Lines 115-117, it suggests that women self-reported adhering to the OGTT process. Suggest saying, 'The majority of women reported consuming'. Self-reporting adherence to treatment regimens is not a reliable measure.

Line 161: As above, suggest rewording 'reported not drinking'.

Qualitative analysis:

Suggest including sub-themes that appear in tables under each heading (reassurance and convenience), with a brief description of each. Longer in-text quotes can be included in tables and more time can be spent describing each sub-theme.

Discussion:

Line 299: What do these findings mean for other contexts, like low- and middle-income countries which also have large burdens of diabetes?

6. PLOS authors have the option to publish the peer review history of their article (what does this mean?). If published, this will include your full peer review and any attached files.

Reviewer #1: **Yes: **Asma Ahmed MRCP(UK), FRCP(London)

Reviewer #2: No

---

## [Author Response · Author response to Decision Letter 0]

19 Jul 2023

Thank you for the opportunity to revise our submission. Please see the detailed response to the reviewers’ comments below. I hope you will now consider it suitable for publication. 

Many thanks,

Claire Meek

---

## [Decision Letter · Decision Letter 1]

17 Aug 2023

PONE-D-23-07848R1Feasibility and acceptability of continuous glucose monitoring in pregnancy for the diagnosis of gestational diabetes: a single-centre prospective mixed methods studyPLOS ONE

Dear Dr. Meek,

Thank you for submitting your manuscript to PLOS ONE. After careful consideration, we feel that it has merit but does not fully meet PLOS ONE’s publication criteria as it currently stands. Therefore, we invite you to submit a revised version of the manuscript that addresses the points raised during the review process.

We look forward to receiving your revised manuscript.

Kind regards,

Graeme Hoddinott, Ph.D

Academic Editor

PLOS ONE

Journal Requirements:

Reviewers' comments:

Reviewer's Responses to Questions

**Comments to the Author**

1. If the authors have adequately addressed your comments raised in a previous round of review and you feel that this manuscript is now acceptable for publication, you may indicate that here to bypass the “Comments to the Author” section, enter your conflict of interest statement in the “Confidential to Editor” section, and submit your "Accept" recommendation.

Reviewer #2: All comments have been addressed

2. Is the manuscript technically sound, and do the data support the conclusions?

Reviewer #2: Yes

3. Has the statistical analysis been performed appropriately and rigorously? 

Reviewer #2: I Don't Know

4. Have the authors made all data underlying the findings in their manuscript fully available?

Reviewer #2: Yes

5. Is the manuscript presented in an intelligible fashion and written in standard English?

Reviewer #2: Yes

6. Review Comments to the Author

Reviewer #2: (No Response)

7. PLOS authors have the option to publish the peer review history of their article (what does this mean?). If published, this will include your full peer review and any attached files.

Reviewer #2: No

---

## [Editor Report · Decision Letter 2]

12 Sep 2023

Feasibility and acceptability of continuous glucose monitoring in pregnancy for the diagnosis of gestational diabetes: a single-centre prospective mixed methods study

PONE-D-23-07848R2

Dear Dr. Meek,

We’re pleased to inform you that your manuscript has been judged scientifically suitable for publication and will be formally accepted for publication once it meets all outstanding technical requirements.

Kind regards,

Graeme Hoddinott, Ph.D

Academic Editor

PLOS ONE
---

## [Editor Report · Acceptance letter]

18 Sep 2023

PONE-D-23-07848R2 

Feasibility and acceptability of continuous glucose monitoring in pregnancy for the diagnosis of gestational diabetes: a single-centre prospective mixed methods study.  

Dear Dr. Meek:

I'm pleased to inform you that your manuscript has been deemed suitable for publication in PLOS ONE. Congratulations! Your manuscript is now with our production department. 

Kind regards, 

on behalf of

Dr. Graeme Hoddinott 

Academic Editor

PLOS ONE